# Land-Use/Land Cover Changes Contribute to Land Surface Temperature: A Case Study of the Upper Indus Basin of Pakistan

**Akhtar Rehman [1], Jun Qin [1,\*], Amjad Pervez [2], Muhammad Sadiq Khan [3], Siddique Ullah [4], Khalid Ahmad [4] and Nazir Ur Rehman [5]**

[1]  School of Environmental Studies, University of Geosciences, Wuhan 430076, China; akhtarktk17@gmail.com
[2]  School of Traffic and Transportation Engineering, Central South University, Changsha 410075, China; amjadpervez04@csu.edu.cn
[3]  State Key Laboratory of Urban and Regional Ecology, Research Center for Eco-Environmental Sciences, University of Chinese Academy of Sciences, Beijing 100085, China; khan_st@rcees.ac.cn
[4]  Department of Environmental Sciences, COMSATS University Islamabad, Abbottabad 22060, Pakistan; siddiqullah142@gmail.com (S.U.); khalid.taxonomist@gmail.com (K.A.)
[5]  Department of Geology, Khushal Khan Khattak University, Karak 27200, Pakistan; nazirktk@yahoo.com
[\*]  Correspondence: qinjun@cug.edu.cn

**Abstract:** Land-use/land cover (LULC) changes have an impact on land surface temperature (LST) at the local, regional, and global scales. To simulate the LULC and LST changes of the environmentally important area of northern Pakistan, this research focused on spatio-temporal LULC and associated LST changes since 1987 and made predictions to 2047. We classified LULC from Landsat TM and ETM data, using the maximum probability supervised categorization approach. LST was retrieved using the Radiative Transfer Equation (RTE) methodology. Furthermore, we simulated LULC using the integrated approaches of Cellular Automata (CA) and Weighted Evidence (WE) and used a regression model to predict LST. The built-up areas and vegetation have increased by 2.1% and 11% due to a decline in the barren land by −8.5% during the last 30 years. The LULC is expected to increase, particularly the built-up and vegetation classes by 2.74% and 13.66%, respectively, and the barren land would decline by −4.2% by 2047. Consequently, the higher LST classes (i.e., 27 °C to <30 °C and ≥30 °C) soared up by about 25.18% and 34.26%, respectively, during the study period, which would further expand to 30.19% and 14.97% by 2047. The lower LST class (i.e., 12 °C to <21 °C) indicated a downtrend of about −41.29% and would further decrease to −3.13% in the next 30 years. The study findings are useful for planning and management, especially for climatologists, land-use planners, and researchers in sustainable land use with rapid urbanization.

**Keywords:** land-use/land cover; land surface temperature; simulation; sustainable land-use planning

## 1. Introduction

The rapid growth of urbanization is one of the dominant and noticeable man-made changes in the world [1–3]. More so than natural forces, anthropogenic factors are continually involved in the energy transition from the Earth's surface to the atmosphere via LULC changes [4] that impact LST. The fundamental effect of urbanization is a fast natural landscape change caused by anthropogenic LULC, which results in various environmental and ecological losses, such as urban heat islands (UHIs) [5,6]. Rapid urbanization and overpopulation in cities are responsible for LULC changes and rose from 54.6% to 78.3% from 1950 to 2015 (UNDP, 2017). For the first time, in 1833, UHIs were calculated in London in a calm and sunny climate that heavily relies on metropolitan structure and land-use classifications [7,8]. Human activities in the past centuries altered the LULC [9], which had significant impacts on terrestrial ecosystems at local [10], regional [11], and global levels [12], and thus on the environment as a whole [13].

LULC change is constant on the terrestrial surface of the Earth [14,15]. Classification of LULC has permitted us to assess changes in LULC in a less costly and more time-efficient manner than ground-based and traditional methods proposed by prior studies [16–18]. Consequently, understanding the dynamic system has great importance not just for studies on changes in LST and LULC, it also has social, economic, and environmental importance. Moreover, there are various types of LULC [5,7]. Thus, it is necessary to determine the existing LULC contribution accurately to calculate the contribution in changes in LST.

LST is the radiant temperature of the land's surface and is affected by geographical conditions, landscaping structure, land cover, and urban sprawl, according to the existing literature [19–21]. Thus far, several researchers have conducted studies using Landsat data, an appropriate sequential assessment for LST [22]. Remote sensing (RS) applications have monitored LULC changes and LST in the past years via Landsat [23]. There are numerous advantages of RS, such as ease of access to remote areas, provision of unavailable data, cost-effectiveness, less fieldwork, and software base compatibility [18]. Therefore, for the estimation of annual median LULC and LST investigation, we choose Landsat data. Moreover, conventional studies that depend on fieldwork are consuming time, cannot apply to extensive studies, and are highly expensive [18].

Simulation models are highly significant in examining the LULC variations in the future [24,25]. LULC modeling can be done through various methods and algorithms. As far as prediction models for LULC, those widely used are Cellular Automata (CA), Artificial Neural Networks (ANN) [26], and Markov Chains [27]. The appropriate models, i.e., CA-ANN and CA-Stochastic models, can estimate many directional changes to provide accurate findings [11,28]. LST and LULC can be estimated and modeled by using GIS and RS. For urban planners, designers, and climatologists, evaluating LULC alterations and their impact on LST will be a valuable contribution. Former work focused mostly on fluctuations in LULC of the metropolitan and rural mountainous regions of Pakistan [29–32], and very limited work has been done on LULC changes and their influence on related LST in northern Pakistan.

Pakistan's northern mountainous areas are important and famous for their natural resources and beauty. However, for several decades, large numbers of people have migrated from other regions and accommodated themselves there, which alters the LULC and LST characteristics. Currently, the region has become more accessible with the construction of a linking road between China and Pakistan under the China Pakistan Economic Corridor (CPEC) project. Changes in the landscape pattern, LULC changes, and the migration of people have increased in the region, which might bring even greater changes in the LULC and LST. However, the extent of the change and its effect on LULC and LST is not clear because no research work has been done. This lack of information has created a gap for researchers, policymakers, urban planners, and climatologists to devise a sustainable land use plan for the local communities. Therefore, the current study aims to assess the current LULC and LST and predict the future scenario. Such a study could help to mitigate the UHIs of the study area, which can apply to other regions with similar conditions.

## 2. Materials and Methods

### 2.1. Study Area

The proposed research area (District Shangla) is adjacent to the Upper Indus Basin, which is a very sensitive and important region of the country in terms of the biophysical environment, attractive landscape, and natural resources, like water bodies, biodiversity, high mountain chains, and diverse valleys with vegetation cover, agriculture, and pleasant climate. The Shangla is a part of the Malakand division of northern Khyber Pakhtunkhwa Province of Pakistan (34.8872° N, 72.7570° E (Figure 1). The general elevation range is 2000 to 3500 m above sea level. Geographically, Khyber Pakhtunkhwa is divided into two zones, the northern one starts from the mountainous area of Hindu-Kush to the boundary of the Peshawar basin, whereas the southern part starts from Peshawar to Derajat basin. The Northern Pakhtunkhwa experiencing cold winter with heavy rainfall in the summer

season except for the Peshawar basin border, which is hot during summer and cold in the winter season. The district Shangla is spread over a 1586 km$^2$ area with a population of 757,810 [33] and is famous for lush green vegetation, and tourism. The yearly mean rainfall is 1040 mm, with a total mean temperature of 15.8 °C [34]. It includes high mountains and a narrow valley on the western side of the Himalayan with more than 70% forest cover.

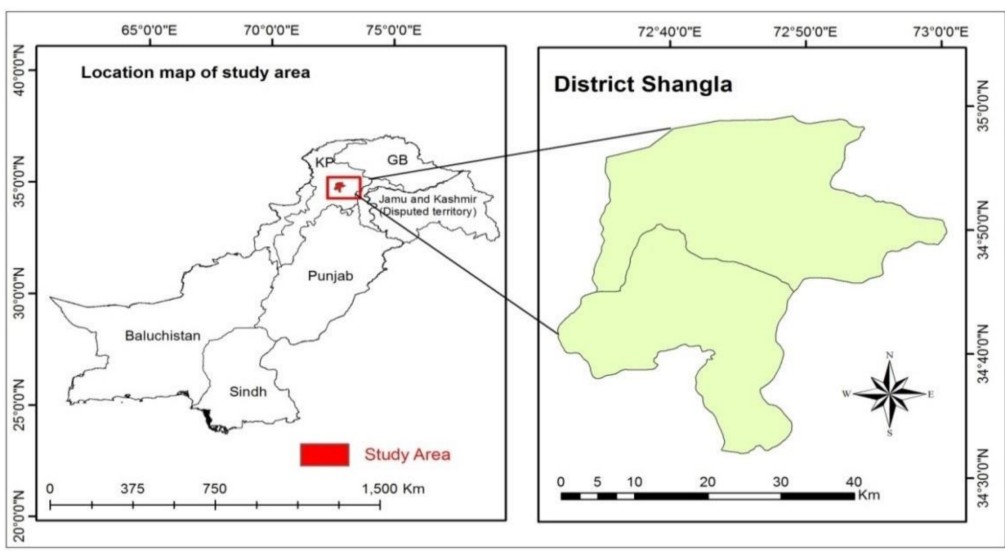

**Figure 1.** Study area map (District Shangla, Pakistan).

## 2.2. Remote Sensing Data Acquirement

The Landsat imageries were acquired from the United State Geological Survey (USGS, NASA) for a period of thirty years with a specified 15 years interval from 1987–2017 to evaluate the changes in LULC and LST. The appropriate imageries were downloaded in May in five days (19–24 May) with cloud-free problems. All data were downloaded at the start of the summer season with the cloud-free option of less than <15% for weather clarity. The images were comparable for environmental variables, i.e., humidity and temperature. The chosen images scene range varies from 6% to 13%. These imageries details are in Table 1.

**Table 1.** List of Landsat images for selected study.

| Satellite | Cloud Cover (%) | Row/Path | Resolution (m) | Acquired Date |
|---|---|---|---|---|
| Landsat 5 | 6% | 150/36 | 30 | 24 May 1987 |
| Landsat 7 | 8% | 150/36 | 30 | 19 May 2002 |
| Landsat 8 | 13% | 150/36 | 30 | 20 May 2017 |

## 2.3. Remote Sensing Data Processing

The pre-processing steps were performed for the satellite data before the LULC and LST classification and retrieval. Further processing included radiometric calibration, atmospherical correction, and line removal in ArcMap. The LULC maps were generated for the years 1987, 2002, and 2017, and consequent LST maps were developed from the thermal bands.

## 2.4. LULC Accuracy Assessment and Classification

For LULC classification, Landsat images were classified using the Support Vector Machine (SVM) technique in software ENVI 5.3 version [35]. Baseline data were obtained via 40 preparation samples for each land-use type, as well as reference locations, which were superimposed on a high-resolution satellite image using Google Earth. All the spatio-spectral profiles and auxiliary information were used for the correctness of training sites.

The correctness of classification was quantified with the kappa coefficient. The overall correctness was gained by applying the uncertainty matrix that is a commonly applied tool for this purpose [36]. The overall methodology of LULC and LST is given in Figure 2.

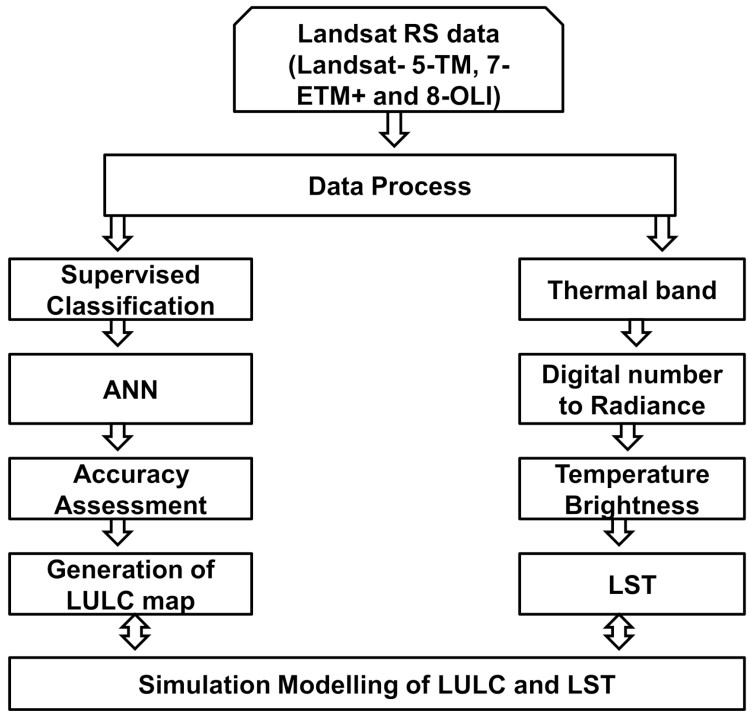

**Figure 2.** Flow chart methodology of the research study.

*2.5. LST Estimation*

Using conventional techniques of proposed RTE, the LST was extracted from the thermal bands of Landsat satellite imageries from both TM and ETM [37]. The Normalized Difference Vegetation Index (NDVI), Vegetation Percentage (PV), and land surface emissivity were used in the RTE methodology (LSE). Equation (1) was used to calculate the NDVI.

$$NDVI = \frac{NIR + RED}{NIR + RED} \tag{1}$$

Here, NIR stands for near-infrared band (band 5) and RED denotes the red band (band 4) in Landsat-8 OLI, whereas bands 4 and 3 show the NIR and RED in Landsat-5 (0.64–0.67 mm) while band 4 (0.85–0.88 mm) in Lansat-8 OLI are nearly constant with band 4 (0.77–0.90 mm) and bands 3 (0.63–0.69 mm) in Landsat-5 and applied for the similar findings.

Equation (2) was used to calculate PV, which was based on the NDVImin and NDVImax standards.

$$PV = \left( \frac{(NDVI - NDVImin)}{NDVImax - NDVImin} \right)^2 \tag{2}$$

For retrieval of LST, assessment of LSE is mandatory, which is a proportion factor, and balances black body radiance (Planck's Law) to forecast emitted radiance [38]. Equation (3) was used to measure LSE:

$$LSEBi = \varepsilon_s (1 - FVC) + \varepsilon_v FVC \tag{3}$$

Here, Ev and Es denote foliage and soil emissivity standards, *Bi* denotes the band number, and C denotes surface roughness (C=0 for smooth surfaces) as a continuous value of 0.005. For band 10, Es and Ev were 0.971 and 0.987, respectively, whereas, for band 11, Es and Ev were 0.977 and 0.989 [39]. An effective technique for retrieving LST from Landsat TM and OLI data was utilized to develop the temperature vividness of the thermal band at

the sensor level. In addition, the temperature brightness from thermal bands includes a radiance evaluation based on the Digital Number (DN) impact. The radiance was evaluated using the National Aeronautics and Space Administration's (NASA) Equation (4), which was obtained from the DN value of Landsat information.

$$Li = RADIANCEMULT_{Bi} \times DN + RADIANCEADD_{Bi} \tag{4}$$

The Li is the spectral radiance of the sensor (m W cm$^{-2}$sr$^{-1}$ μm$^{-1}$), RADIANCEMULT and RADIANCEADD are constant bands, existing in the header file. We analyze the temperature vividness from Equation (5).

$$Ts_{Bi} = \frac{K2_{Bi}}{log\left(\left(\frac{K1_{Bi}}{Li}\right) + 1\right)} - 273.15 \tag{5}$$

Here, Ts indicates temperature brightness at the satellite of band I in Kelvin while K1 and K2 are constants. This technique was designated by USGS to calculate temperature brightness. To estimate LST in Celcius from Kelvin, we deducted 273.15 from the outcomes [40]. Assessing LST, top of atmospheric (TOA) spectral radiance must be accurate to get surface spectral radiance as atmospheric effects are very important for the studies of temperature [41]. In the current research work, we applied a standard RTE method proposed by [37] shown in Equation (6).

$$LST_{RTEBi} = EiTi + ((1 - Ei)\ Downwelling) + Upwelling \tag{6}$$

Here, E indicates surface emissivity of band 1, Ti is a spectral radiance, and downwelling and upwelling, are the path radiance. For the downwelling and upwelling calculations, the MORTAN 5.0 radioactive transfer code executes applying 1976 standard US atmospheric profiles choosing the Urban Aerosol Model [42]. Ground radiance (Ti) is calculated as follows using Planck's law:

$$Ti = \frac{C_1}{Wavelength_{Bi}^5\left(exp\left(\frac{C_2}{Wavelength_{Bi}}\ Ts\right) - 1\right)} \tag{7}$$

Here, C1 and C2 are Planck's radiation constants (C1 is 1.19104 × 108 Wμm4 m$^{-2}$ sr$^{-1}$ and C2 is 14,387.7 μm k), wavelength indicates the wavelength of bands (band 10 = 10.602 and band 11 = 12.511), and Ts is the surface temperature generated from Equation (5). The technique of RTE is for a single band and that is why used for both (10 and 11 bands). The results of Equation (6) for every band were put in Equation (7) to measure mean LST. We applied thermal band 6 of Landsat-5 and thermal band 10 of the Landsat-8 OLI images because of supplementary relevancies in the investigation of LST results. Band 11 was disregarded because of LST evaluation error, and the motive is the effect of absorption of water vapor and sensitivity, as proposed in an earlier study [39].

*2.6. Detection of Relative LST Change*

To compare the influence of LULC on the urban thermal environment, the relative LST was obtained for the years 1987, 2002, and 2017. The involvement of RLST from LULC (increase/decrease) alteration is obtained from the mean LST of the planned study area and through each pixel value through Equation (8).

$$RLST\ Ij = LSTij - LSTi\ mean \tag{8}$$

Here, RLST ij displays the temperature of pixel j of class i; LST ij represents the temperature of cell j of class i, and LST i shows the mean value of LST for the urban landscape. If RLSTji > 0, the pixel indicates positive involvement of transfiguration of LULC, and if RLST ji < 0, then it is a negative impact on the thermal environment [43,44].

*2.7. Classification of Temperature Zones*

For easy interpretation, LST was categorized into different groups based on the ranges, such as; <12 °C, 12 °C to <21 °C, 21 °C to <24 °C, 24 °C to <27 °C, 27 °C to <30 °C, ≥30 °C. Estimates were then made to assess the proportion of an area for each group. The highest temperature range was equal to or more than 30 °C while the lowest was less than 12 °C using Equation (9).

$$LSTs = \frac{LST - LSTu}{LST\Omega} \tag{9}$$

*2.8. LULC Simulation for the Next Thirty Years (2047)*

Prediction models were used to predict the future up to the year 2047, based on the classified images for the year 2017 [45]. Firstly, it modeled the ANN-transformation potential matrix, and then forecast possible changes in LULC using the CA model to predict future LULC in the MOLUSCE tool in QGIS. Land-use models like the CA model, which cover both the static and dynamic side of LULC shifts, were used to forecast land cover changes due to high accuracy [27]. The forecasting depends on different variables; like previous LULC changes calculated from 2002 and 2017 images of Landsat and space from the roads, height, and slope.

The road distance was assessed by using vector data in the study area and the Euclidian distance function in ArcMap software. Once it was modeled using the logistic regression technique, then the CA model was used to predict the upcoming LULC maps for the year 2047 using the MOLUSCE tool in QGIS software. To make sure that the model is suitable in forecasting LULC change for a particular projected year, it was validated by applying baseline available records. Hence, the CA model was validated for correctness to simulate LULC for the year 2017, which was compared to assess LULC (from Landsat data) of a similar year in the study area. Moreover, the QGIS–MULUSCE validation module was applied to measure overall Kappa coefficients and percentage of accuracy between classified and simulated LULC map of 2017. Thus, the correctness of the CA model was assessed by two approaches described beyond using it for the upcoming simulations.

*2.9. LST Simulation for the Year 2047*

The major issue for public managers in the contemporary situation of global warming is growing LST in metropolitan zones [46]. A multilayer feed-forward back-propagation Artificial Neural Network approach in MATLAB software [47] was utilized to simulate and predict future LST change by applying the previous trends in a particular region. The Multi-Layer Perceptron (MLP) neural network responds automatically to network parameters and how they should be changed to improve the network's model. The MLP algorithm is based on the error accuracy-learning concept. In MLP, when the network gets a series and its processes, it probably provides less accurate random output.

The LST prediction in the study relied on the LST information trends from 1987 to 2017. The research region was further subdivided into spatial grids of 500 m × 500 m to create a sample point range of spatial units applying QGIS software. This grid size was preferred because there is a smaller area range where the features of a single point can have a significant impact. For LST prediction, the sample record was used to build a Neural Network in MATLAB. Moreover, latitude and longitude of the distinct sample spatial units were added to increase the network model's efficiency. The more input parameters, the greater the network model's efficiency.

LST's forecasts included network construction, network training, network performance evaluation, and prediction. Mean Square Error (MSE) and correlation coefficient (R) standards measured the network confidence. The study of regression yields measurements of how well the target data set explained variation in the output findings. When the value is 1, the output and target data sets are perfectly connected. Getting the value of one, on the other hand, is quite difficult. Before implementing the network, the Graphic User Interface

(GUI) was created to assess the performance indication. Once the performance indicator was okay then these were saved for prediction.

Based on MSE and R, various hidden layers were selected. The hidden layers are important because they allow the network to display non-linear behavior, which affects the outcomes. Three hidden layers were chosen for the recent study. The first learning rate (μ) during the research was fixed at 0.1 and the decay rate (β) was utilized to keep it constant. The standard decay rate lies in the range of 0–1 (0 < β). The decay rate of 0.9 was used to update the learning rate, which means that if the error function between the current and previous iteration is increasing, the β upgrades the learning rate μ by division, but when the error function is decreasing, it multiplies it to retreat the μ.

### 2.10. NDVI and LST Correlation

The LULC indices were updated for the LST projection of the year 2047. The various indices used to estimate the change in LST were NDBaI, NDBI, NDVI, and UI (Table 2). The bivariate analysis between indices and LST was performed to identify the indices that have a strong association with LST based on the LULC indices. However, index NDVI was chosen to indicate the significant correlation with LST and their values set as ($p < 0.05$), R and RMSE values were 0.485 and 2.942. A regression model with selected index NDVI and LST was used to generate the regression equation.

**Table 2.** LULC indices for simulation of LST, 2047.

| Indices | Equation (Landsat 5 TM and 7 ETM+) | Landsat 8 OLI |
|---|---|---|
| NDVI | B4 − B3/B4 + B3 | B5 − B4/B5 + B4 |
| UI | B7 − B4/B7+B4 | B7 − B5/B7 + B5 |
| NDBal | B5 − B6/B5 + B6 | B6 − B10/B6 + B10 |
| NDBI | B5 − B4/B5 + B4 | B6 − B5/B6 + B5 |

B3, 4,5,6,7,10 indicate bands of Landsat data.

The subsequent equation was used for NDVI and LST for the year 2017.

$$\text{LST} = 30.10 + (2.14 \times \text{NDVI}) \quad \text{LST} = 24.24 + (12.20 \times \text{NDVI}) \tag{10}$$

Since CA-ANN model predicts classes, the NDVI index was reclassified so that they could predict <12, 12–<21, 21–<24, 24–<27, 27–<30, ≥30 °C. The preferred NDVI index of 2017 was modeled using CA-ANN to forecast the alteration between 2017 to 2047. Finally, the regression equation and simulated NDVI index were used to model LST for the year 2047.

## 3. Results

We recognized the changes in LULC and LST patterns in the last 30 years (1987–2017) and then modeled these data for the next thirty years (i.e., 2047).

### 3.1. Past Changes in LULC

The results show that the built-up area and vegetation classes were 4.4% and 55.1% in the year 1987 and increased to 6.5% and 66.1%, respectively, in 2017 (Figures 3 and 4). On the other hand, barren land has decreased from 23.7% to 15.2% since 1987. Various elements may contribute to the class change, e.g., an increase in the built-up class is associated with urbanization.

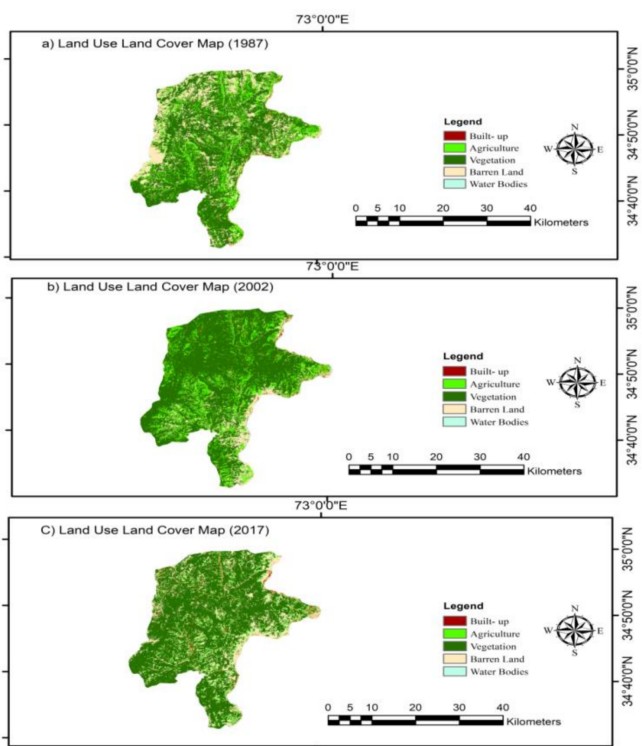

**Figure 3.** LULC maps of Landsat images for the years (**a**) LULC (1987), (**b**) LULC (2002), (**c**) LULC (2017).

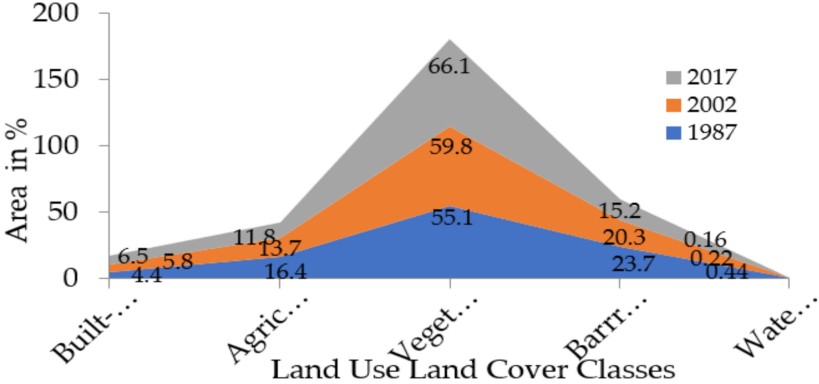

**Figure 4.** Changes in LULC classes in the area (%) during the study period.

### 3.2. Past Changes in Land Surface Temperature (1987–2017)

The classified LST map was grouped as, <12; 12 to <21; 21 to <24; 24 to <27; 27 to <30; and ≥30 (Table 3; Figure 5). The maximum temperature level was set equal to or above 30 °C, whereas the minimum temperature was fixed at less than 12 °C, which demonstrates the changes in these groups during the years 1987, 2002, and 2017. The temperature zone of higher classes in series of 27 °C to 30 °C and equal to or above 30 °C were 1.40% and 0.3% in the year 1987, which increased to 26.58% and 34.29% in the year 2017. The lower group in series of 12 °C to <21 °C was declined from 44.43% to 3.14% during the study period. The majority of lower temperature areas converted into upper-temperature zone due to changes in climate and urbanization showed rising drift in the study area's LST. The current study showed that built-up areas have higher LST than others.

**Table 3.** Alteration in LST in (%) area during the year 1987, 2002, and 2017.

| Temperature Range (°C) | Area (%) 1987 | Area (%) 2002 | Area (%) 2017 |
|:---:|:---:|:---:|:---:|
| <12 | 15.94 | 31.65 | 00.26 |
| 12 to <21 | 44.43 | 34.84 | 03.14 |
| 21 to <24 | 28.65 | 17.55 | 12.06 |
| 24 to <27 | 09.54 | 10.34 | 23.68 |
| 27 to <30 | 01.40 | 03.45 | 26.58 |
| ≥30 | 00.03 | 02.16 | 34.29 |

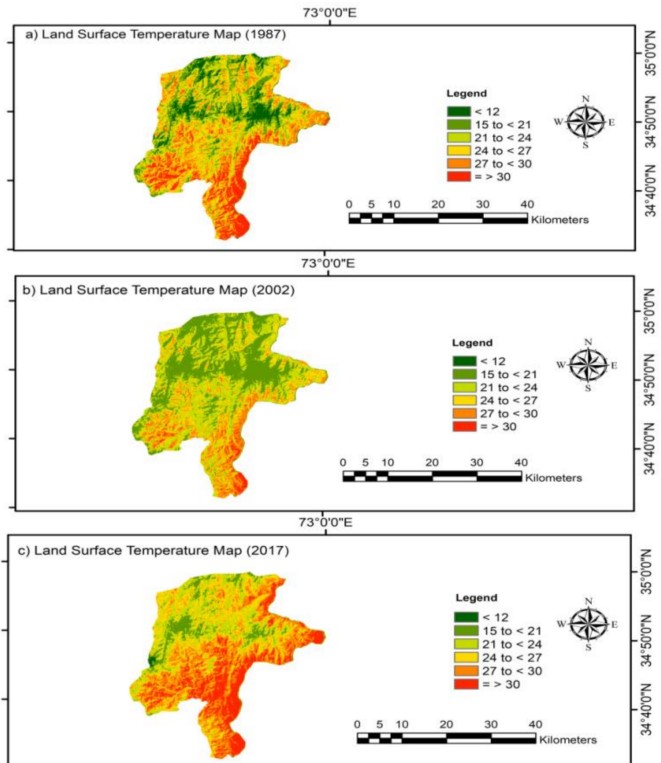

**Figure 5.** LST change based on Landsat thermal bands for the years (**a**) LST (1987), (**b**) LST (2002), (**c**) LST (2017).

### 3.3. LST and NDVI Correlation and Regression

The relationship between NDVI and LST was assessed by Pearson's correlation value (r) and found a negative correlation between NDVI and LST between the year 1987 and 2017, and the highest correlation value (r) was recorded −0.60 for the year 2017 (Figure 6; Table 4). As the areas have maximum vegetation cover, so the downward trend is strong.

**Table 4.** Correlation and regression analysis of NDVI and LST from 1987 to 2017.

| Years | Correlation | R Square | Intercept Value |
|:---:|:---:|:---:|:---:|
| 1987 | −0.26 | 0.07 | 15.01 |
| 2002 | −0.54 | 0.29 | 17.11 |
| 2017 | −0.60 | 0.36 | 29.25 |

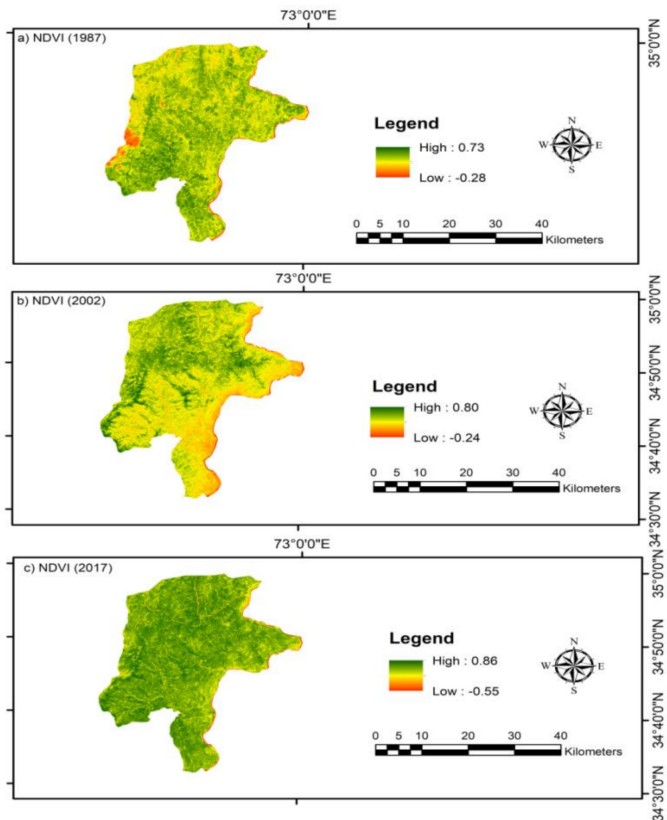

**Figure 6.** Alteration in NDVI maps for the years; (**a**) NDVI (1987), (**b**) NDVI (2002), (**c**) NDVI (2017).

### 3.4. Simulation of the Future LULC

The results show LULC alteration during the year 1987 to 2017 and for the simulated year 2047. The forecasted result indicated that the barren land has decreased (4.2%) and the built-up and vegetation cover increased by 2.74% and 13.66%, respectively (Figure 7).

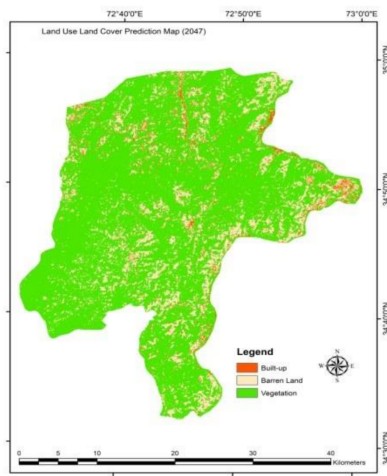

**Figure 7.** LULC prediction for the year 2047.

### 3.5. Simulation of the Future LST

The remarkable LST changes consequent to LULC types in the area during the study period (1987–2017). The simulated LST estimation between the projected and acquired LST indicates a strong connection that proved the accuracy prediction. The forecasted results designated the rising drift of 30.19% of the area in the higher temperature class (i.e., 27 to

<30 °C) while the declining tendency of 3.13% in the area of the lower class range (i.e., 12 to <21 °C) (Figure 8).

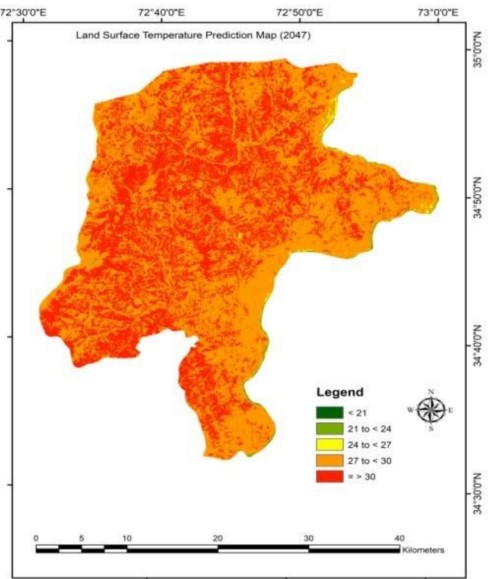

**Figure 8.** Simulation of LST map for the year 2047.

## 4. Discussion

### 4.1. Past Changes in LULC

The migration of people from adjoining areas and the dissatisfied situation after the year 2005 earthquake in the allied western and eastern Himalayan regions seems the main cause of urbanization. The finding of the present study supports the result of the past studies [48,49]. The geographical, political, economic, technological, and ecological shifts seem to be accountable for urbanization. The LULC change is the main issue, which is the driving factor of ecological changes. Every year, people's migration to the cities leads to an abrupt shift in the urban area's ecosystem, biodiversity, natural landscape, geography, and environment [50]. Moreover, the overall vegetation level in the study area indicates a rising pattern. The reasons are a high altitude that attributes to replacing forest cover and the project "Billion Tree Tsunami" planned by the KPK government (IUCN, 2017). According to a previous study [51], foliage in the study area might be increased because of huge plantations inaugurated by the forest department and the approach of the local community was doing cultivation at forestland. Additionally, substantial plantations have been taken out in nearby areas of the district of Shangla [20].

### 4.2. Past Changes in Land Surface Temperature (1987–2017)

The majority of lower temperature areas converted into upper-temperature zone due to changes in climate and built-up expansion showed rising drift in the study area's LST. The current study shows that built-up areas have higher LST than others. Similar results were reported by different researchers [52,53], revealing that the Himalayan plateau is an important point of rainfall to discharge heat, as variation in temperature is formed, linking the air at elevated and low elevation, and its impact is less at lower altitudinal parts like our study area [54]. The hot insulation and damp air between the subtropics and the polar areas is another important feature that devastates the climate of the Himalayan region. The current research focus is on changes in LULC and LST pattern (rather than climate changes) if the local heat influences the air temperature at the Indus Basin area, which alters the temperature and moist the air in the boundary area [55].

*4.3. LST and NDVI Correlation and Regression*

As the areas have maximum vegetation cover, so the downward trend is strong. Such a trend is supported by studies of the correlation between NDVI and LST in Shangai, China [56], and other studies [57,58]. Several factors might affect the relationship between NDVI and LST such as time of the day, annual season, changes in vegetation, and climate change. For example, the vegetation friction had a stronger negative correlation with LST as compared to other classes [59], and NDVI and LST have a negative correlation in summer while positive in the winter season for the same year [60]. Similarly, our study area is located at the Upper Indus Basin and humid tropical zone that may be the primary reason for a negative correlation between NDVI and LST.

*4.4. Simulation of the Future LULC*

This research work found LULC alteration which is supported by a previous study in Beijing [61] that overpopulation is the key reason for enhancing these alterations. Prior management is required to look after the study area from harsh changes, as 49.91% of inhabitants are accommodated in urbanized areas and will probably reach 60% of the total population of the world by 2030. Moreover, the number of megacities will arrive at 100% by 2025 [62].

*4.5. Simulation of the Future LST*

The forecasted results show the increasing drift (30.19%) in the higher temperature class (i.e., 27 to <30 °C) while the declining tendency (3.13%) in the lower-class range (i.e., 12 to <21 °C) (Figure 7). Numerous factors such as greenhouse gases, global warming, and alterations in surface characteristics are involved that eventually influence the area's LST [63,64]. Similarly, the enlarged metropolitan areas may also be the reason for temperature expansion. The expansion in the LST area also influences the thermal power of land cover, which leads to UHIs from the land surface alteration that courteously traps heat from manmade sources [8]. This phenomenon particularly enhances the urban temperature [65], which is the key ecological problem for humans and biodiversity [66]. The warming trend in the South Asian area is expected to be greater than the average global temperature (IPCC, 2014). However, the effect of global warming could be mitigated by vegetation cover near the building site [67,68].

**5. Conclusions**

The LST connection with LULC changes is a well-known phenomenon across the globe. Therefore, the proposed study aimed to investigate the changes in LST and LULC for the past 30 years and to predict for the year 2047 in the Upper Indus Basin area of Pakistan. The results can be concluded as:

1.  The transformation of natural surfaces into artificial surfaces induces changes in LST. A remarkable LULC change was shown in the built-up and vegetation areas, which were enlarged by 2.1% and 11% and have higher surface temperatures.
2.  Increasing green cover can contribute to the mitigation of UHIs, while the increase in the barren land and built areas support the UHIs effect.
3.  LST showed a rising trend due to changes in climate and urban warming. It was also observed that the lower temperature zones are moving to high-temperature zones, which could lead to UHIs configuration.
4.  The simulation model indicates that LULC and LST pattern increases, and decrease would be in the same trends preceded as the past except for natural disasters.

In general, the study findings provide major insights and a base of concern for landscape managers to take action for monitoring the unplanned built-up expansion and associated patterns of UHIs. The most important concern is about the newly constructed link road under the CPEC project, which passes from the areas and may have an impact on the LULC change in the region. Mitigation measures need to be timely adopted. The limitations of the present study are unavailability/inaccessibility of the high-resolution

images, which may affect the finding's accuracy, and more intensive studies with 5-year intervals for a better understanding of the thermal dynamics and its relationship with LULC change. Additional research is needed to overcome the above shortcomings and use a different and multiple-model approach for further clarifications.

**Author Contributions:** Conceptualization, A.R.; methodology, J.Q., A.R., M.S.K. and S.U. software, A.R. and S.U.; validation, M.S.K.; S.U. and K.A.; formal analysis, A.R., A.P. and S.U.; investigation, N.U.R. and A.R.; resources, J.Q. and S.U.; data curation, A.R. and S.U.; writing—original draft preparation, A.R.; writing—review and editing, M.S.K., A.P., N.U.R., A.R.; visualization, A.R.; supervision, K.A., J.Q.; project administration, J.Q.; funding acquisition, J.Q. All authors have read and agreed to the published version of the manuscript.

**Funding:** The project is self-designated, supported, and funded by the National Key R & D Program of China (2018YFA0605603).

**Institutional Review Board Statement:** Not applicable.

**Informed Consent Statement:** Not applicable.

**Data Availability Statement:** The Landsat satellite data for the proposed study was freely available on the online portal of the United States Geological Survey (USGS-NASA) website.

**Acknowledgments:** The authors are grateful to the China University of Geosciences (Wuhan) and the United States Geological Survey (USGS-NASA) for making Landsat satellite data publicly available. We also extend our gratitude to the anonymous referees of the manuscript for their useful annotations and helpful recommendations.

**Conflicts of Interest:** The authors declare no conflict of interest.

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
