# Peer review of "Land-Use/Land Cover Changes Contribute to Land Surface Temperature: A Case Study of the Upper Indus Basin of Pakistan"

_sustainability, doi:10.3390/su14020934_

Round 1
Reviewer 1 Report
The topic of the paper is interesting and up-to-date. The literature basis is good and the research methodology is well described. The paper is well written and readable.
The paper itself has some flaws worth improving:
- Describe research gap of the paper and research questions.
- Please describe the links between the research gap and the goal of the paper and research question. Write why the paper is important. What is the main contribution of the paper to the field?
- Please add some limitation of the study in the conclusion.
- Describe social impact of the paper and what research can be done in the future on the basis of the paper.
Author Response
Response to Reviewer-1 Comments
Thank you for the valuable comments and time on our manuscript. The entire manuscript is updated according to the given comments.
Note: The line numbers of the revised manuscript are updated based on the same statement of the original version of the manuscript. All suggested changes in the manuscript are highlighted with yellow color text in the entire manuscript.
Reviewer-1
Describe research gap of the paper and research questions.
Answer: Line-93-99
Research Gap:
Pakistan Northern mountainous areas are important and famous for their natural resources and beauty. However, large numbers of people migrated from other regions and accommodated the area for several decades, which alters LULC and LST. Currently, the region become more accessible with the construction of a link road between China and Pakistan under the China Pakistan Economic Corridor (CPEC) project. Changes in landscape pattern, LULC changes, and migration of people have expectedly increased in the region, which might bring more vicissitudes in the LULC and LST. This lack of information created a gap for researchers, policymakers, urban planners, and climatologists to devise a sustainable land use plan for the local communities. Therefore, the current study was aimed to assess the current LULC and LST and predict its future scenario. Such a study could help to mitigate the UHIs of the study area which can apply to other regions with similar conditions.
Research Question:
Changes in landscape pattern, LULC changes, and migration of people have expectedly increased in the region, which might bring more vicissitudes in the LULC and LST. But what is the extent of change and its effect on LULC and LST, is not clear because no research work is done.
Please describe the links between the research gap and the goal of the paper and the research question. Write why the paper is important. What is the main contribution of the paper to the field?
Answer: All are interconnected and described as “Changes in landscape pattern, LULC changes and migration of people increased in the region, which might bring more vicissitudes in the LULC and LST. But what is the extent of change and its effect on LULC and LST, isn’t clear because no research work is done. This lack of information created a gap for researchers, policymakers, urban planners, and climatologists to devise a sustainable land use plan for the local communities. Therefore, the current study was aimed to assess the current LULC and LST and predict its future scenario. Such study could help to mitigate the UHIs of the study area which can apply to other regions with similar conditions”.
Importance of the study area:
The proposed research area (Shangla) is adjacent to the Upper Indus Basin, which is a very sensitive and important region of the country in terms of the biophysical environment, attractive landscape, and natural resources, like water bodies, biodiversity, high mountain chains, diverse valleys with vegetation cover, agriculture, and pleasant climate. Please see lines 102-105.
Please add some limitations of the study in the conclusion.
Answer: The Limitations of the present study are unavailability/inaccessibility of the high-resolution images, which may affect the findings accuracy, and more intensive studies with 5 years intervals for a better understanding of the thermal dynamics and its relationship with LULC changes. Please see lines 446-451.
Describe the social impact of the paper and what research can be done in the future based on the paper.
Answer: This lack of information created a gap for researchers, policymakers, urban planners, and climatologists to devise a sustainable land use plan for the local communities. The study findings provide major insights and a base of concern for landscape managers to take action for monitoring the unplanned built-up expansion and associated patterns of UHIs.
Additional research is needed to overcome the shortcomings of this study and use different and multiple-model approaches for further clarifications.

Reviewer 2 Report
This paper is presented as an interesting and current study, as the dynamics of land use in the context of current climate change arouses interest in several fields of activity, from those related to agriculture to those related to health, economy, biodiversity and environmental protection.
I appreciate the approach of the subject, which combines remote sensing methods (LULC, LST) with statistical methods in order to create an overview of future scenarios in your area of study.
However, I believe that for a better understanding of this work some modifications are necessary, which I will highlight below. My analysis was done by chapters, but my remarks will only focus on part of them.
Introduction: This chapter respects the scientific character of such a paper, the general topic of the article and of being outlined and explained by a sufficient number of quotations (citations).
Materials and methods: This chapter is representative of this paper and is treated in detail. I appreciate the details mentioned in order to obtain both LST and LULC, but although those steps are well explained, they are also difficult to follow.
- I recommend making a workflow diagram that highlights the entire process step by step.
- The document contains many abbreviations. Therefore, I recommend to add a list of abbreviations. Please check if all abbreviations are necessary. The less abbreviations, the easier it is to read the text
- Try to arrange table 2. in the version I received it is not arranged and is more difficult to read.
- Please redo the cartographic materials (fig. 2, 3, 4). Try to enlarge them and rearrange the elements (The legend and the north sign are as big as the map, and the map should be the center of attention). You can also add a histogram with the surfaces / class for each map in the case of figures 2 and 3 to see the evolution more easily.
Discussions: in this chapter the results are analyzed and compared with the results of other similar studies, which completes the overall picture of the relationship between LULC and LST.
Conclusions: This last chapter treats very well, in a compact and easy to understand way the complex study that has been done. From my point of view, nothing should be changed in this regard.
In conclusion, the study is a well-developed one and from my own perspective minor changes should be made only to the Material and methods and chapters, as I wrote above.
Author Response
Response to Reviewer-1 Comments
Thank you for the valuable comments and time on our manuscript. The entire manuscript is updated according to the given comments.
Note: The line numbers of the revised manuscript are updated based on the same statement of the original version of the manuscript. All suggested changes in the manuscript are highlighted with yellow color text in the entire manuscript.
Reviewers-2
Materials and methods: This chapter is representative of this paper and is treated in detail. I appreciate the details mentioned to obtain both LST and LULC, but although those steps are well explained, they are also difficult to follow.
I recommend making a workflow diagram that highlights the entire process step by step.
Answer:
Q: The document contains many abbreviations. Therefore, I recommend to add a list of abbreviations. Please check if all abbreviations are necessary. The less abbreviations, the easier it is to read the text
Answer:
LULC: Land Use Land Cover
LST: Land Surface Temperature
TM: Thematic Mapper
ETM: Enhance Thematic Mapper
OLI: Operational Land Imager
RTE: Radiative Transfer Equation
CA: Cellular Automata
WE: Weighted Evidence
UHIs: Urban Heat Islands
RS: Remote Sensing
GIS: Geographic Information System
ANN: Artificial Neural Network
SVM: Support Vector Machine
CPEC: China Pak Economic Corridor
USGS: United State Geological Survey
NASA: National Aeronautic Space Administration
NDVI: Normalized Difference Vegetation Index
PV: Percentage Vegetation
DN: Digital Number
TOA: Top of Atmospheric
MLP:Multi Layer Perceptron
MSE: Mean Square Error
GUI: Graphic User Interface
Try to arrange table 2. in the version I received, it is not arranged and is more difficult to read.
Answer: Table 2 is arranged into readable form.
Please redo the cartographic materials (fig. 3, 4, 5). Try to enlarge them and rearrange the elements (The legend and the north sign are as big as the map, and the map should be the center of attention). You can also add a histogram with the surfaces/class for each map in the case of figures 3 and 4 to see the evolution more easily.
Answer: We did the above amendment in (figures 3,4 and 5) which are available in the manuscript.
Discussions: The results are analyzed and compared with the results of other similar studies, which completes the overall picture of the relationship between LULC and LST.

Round 2
Reviewer 1 Report
Authors implemented my remarks.